# In Vivo Antistress Effects of Synthetic Flavonoids in Mice: Behavioral and Biochemical Approach

**DOI:** 10.3390/molecules27041402

**Published:** 2022-02-18

**Authors:** Mehreen Ghias, Syed Wadood Ali Shah, Fakhria A. Al-Joufi, Mohammad Shoaib, Syed Muhammad Mukarram Shah, Muhammad Naeem Ahmed, Muhammad Zahoor

**Affiliations:** 1Department of Pharmacy, University of Malakand, Dir (Lower), Chakdara 18800, Khyber Pakhtunkhwa, Pakistan; mehreenghias@yahoo.com (M.G.); pharmacistsyed@gmail.com (S.W.A.S.); mohammadshoaib13@yahoo.com (M.S.); 2Department of Pharmacology, College of Pharmacy, Jouf University, Aljouf 72341, Saudi Arabia; faaljoufi@ju.edu.sa; 3Department of Pharmacy, University of Swabi, Swabi 23460, Khyber Pakhtunkhwa, Pakistan; mukaramshah@uoswabi.edu.pk; 4Department of Chemistry, The University of Azad Jammu and Kashmir, Muzaffarabad 13100, Pakistan; drnaeem@ajku.edu.pk; 5Department of Biochemistry, University of Malakand, Dir (Lower), Chakdara 18800, Khyber Pakhtunkhwa, Pakistan

**Keywords:** flavones and flavonols, antioxidant, stress, in vivo study, biomarkers

## Abstract

Natural flavonoids, in addition to some of their synthetic derivatives, are recognized for their remarkable medicinal properties. The present study was designed to investigate the in vitro antioxidant and in vivo antistress effect of synthetic flavonoids (flavones and flavonols) in mice, where stress was induced by injecting acetic acid and physically through swimming immobilization. Among the synthesized flavones (**F1–F6**) and flavonols (**OF1–OF6**), the mono para substituted methoxy containing **F3** and **OF3** exhibited maximum scavenging potential against DPPH (2,2-diphenyl-1-picrylhydrazyl) with IC_50_ of 31.46 ± 1.46 μg/mL and 25.54 ± 1.21 μg/mL, respectively. Minimum antioxidant potential was observed for **F6** and **OF6** with IC_50_ values of 174.24 ± 2.71 μg/mL and 122.33 ± 1.98 μg/mL, respectively, in comparison with tocopherol. The ABTS scavenging activity of all the synthesized flavones and flavonols were significantly higher than observed with DPPH assay, indicating their potency as good antioxidants and the effectiveness of ABTS (2,2′-azino-bis(3-ethylbenzothiazoline-6-sulfonate) assay in evaluating antioxidant potentials of chemical substances. The flavonoids-treated animals showed a significant (* *p* < 0.05, ** *p* < 0.01 and *** *p* < 0.001, *n* = 8) reduction in the number of writhes and an increase in swimming endurance time. Stressful conditions changed plasma glucose, cholesterol and triglyceride levels, which were used as markers when evaluating stress in animal models. The level of these markers was nearly brought to normal when pre-treated with flavones and flavonols (10 mg/kg) for fifteen days in experimental animals. These compounds also considerably reduced the levels of lipid peroxidation (TBARS: Thiobarbituric acid reactive substances), which was significant (* *p* < 0.05, ** *p* < 0.01 and *** *p* < 0.001, *n* = 8) compared to the control group. A significant rise in the level of catalase and SOD (super oxide dismutase) was also observed in the treated groups. Diazepam (2 mg/kg) was used as the standard drug. Additionally, the flavonoids markedly altered the weight of the adrenal glands, spleen and brain in stress-induced mice. The findings of the study suggest that these flavonoids could be used as a remedy for stress and are capable of ameliorating diverse physiological and biochemical alterations associated with stressful conditions. However, further experiments are needed to confirm the observed potentials in other animal models, especially in those with a closer resemblance to humans. Toxicological evaluations are also equally important.

## 1. Introduction

Stress is a feeling or condition experienced in humans when a person become frustrated and angry/nervous. Stress is actually the reaction of the body towards the demands that he faces, and a number of chemical substances are produced as a result of these reactions, collectively called stressors [1]. Various types of stressors target the brain, which is highly sensitive to degenerative conditions induced by stress [2]. It has been reported by the World Health Organization that about 450 million people are suffering from psychiatric disorders; this contributes to the global diseases burden by a share of 12.3% [3]. During stressful conditions such as acute restraint stress (ARS), many cellular events are stimulated and body energy requirements are increased, resulting in increased levels of reactive oxygen species (ROS), causing oxidative stress [4]. These reactive free radicals cause damage to many parts of the body, especially to the CNS (central nervous system), as the brain consumes a high amount of oxygen, has a high content of lipids and relatively less antioxidant enzymes [5]. Likewise, in rodents, restraint stress triggers numerous hormonal, neurochemical, and behavioral dysfunctions that are often associated with an imbalance of the intracellular redox state in the brain. Numerous reports have shown that in rodents, in restraint stress there is an increase in the peroxidation of lipids (leading to ROS production) and a lower level of antioxidant enzymes are encountered in the brain [5,6,7]. The ROS produced can damage the brain tissues in the form of damage to DNA, proteins and lipids, leading to various pathological disorders [4]. 

Various strategies have been reported to reduce stress effectively that are broadly categorized into non-pharmacological and pharmacological methods [8]. The use of various agents (anti-stress) such as benzodiazepines (diazepam), anxiolytics, CNS stimulants (caffeine and amphetamine), and also some anabolic steroids, despite showing considerable anti-stress potentials against various stress models, have not been proven to be effective against the adverse effects of chronic stress on male sexual function, behavior cognition, immunity, pregnancy and lactation. Moreover, toxicity, physical dependence and tolerance to their extended use minimize the clinical utilization of these agents [8,9,10]. Therefore, a safer, cheaper and effective anti-stress drug to manage stress-induced disorders is desperately needed and poses a significant challenge to scientists. Therefore, research has focused on the discovery and development of new, effective drugs of plant synthetic origin, which are normally tested in different animal models. The chemically-induced stress, forced-swimming and immobilization stress models are widely used tests for evaluating stress in animals [11,12]. 

Flavonoids belong to a group of secondary metabolites that are widely present in the plant kingdom and have been recognized for their exciting medicinal actions [13]. Humans usually obtain large amounts of flavonoids; an important group of naturally occurring, ubiquitous bioactive polyphenolics, from plants, which are then utilized as foods [14]. Flavonoids and the food products that contain them have potential positive effects on stress [15]. Among them, the natural flavones and their synthetic derivatives have been recognized to have antioxidant, antitumor, antiallergic, anti-inflammatory, cardioprotective and neuroprotective activities [16]. The antioxidant potentials of flavones render them as a protective agent of oxidative stress damage [13]. Our previous findings have shown that halogen containing flavones are good enzyme inhibitors and antioxidants that could be used as anti-inflammatory and anti-Alzheimer drugs [17,18]. Other interesting findings by our group reveal that synthetic flavone and flavonol derivatives have noteworthy in vivo, in vitro, and ex vivo memory enhancing effects in a scopolamine-induced amnesic model [19]. From the literature review, it can be established that little work has been carried out investigating the biological application of these synthetic flavones and flavonols as anti-stress agents, and extensive research is needed. Additionally, as mentioned before, a safer, cheaper and effective anti-stress agent in the management of stress induced disorders is required. Therefore, the present research was undertaken to synthesize and estimate the effect of flavonoids as antioxidant and antistress agents.

## 2. Results

### 2.1. In Vitro Antioxidant Activity

The significant concentration dependent antioxidant response by flavonols has been produced in comparison to standard (Figure 1).

Among the synthesized flavones (**F1–F6**) and flavonols (**OF1–OF6**), the mono para substituted methoxy containing **F3** and **OF3** produced maximum scavenging property against DPPH with IC_50_ of 31.46 ± 1.46 μg/mL and 25.54 ± 1.21 μg/mL, respectively. Minimum antioxidant potentials were observed for **F6** and **OF6**, with IC_50_ values of 174.24 ± 2.71 μg/mL and 122.33 ± 1.98 μg/mL, respectively, in comparison to tocopherol. The ABTS scavenging activity of flavones (**F1–F6**) and flavonols (**OF1–OF6**) was significantly higher, indicating its potency as a good antioxidant (Figure 1). 

It is evident that mono para substituted methoxy containing flavone (**F3**) and flavonol (**OF3**) showed an excellent response in contrast with other flavones and flavonols. These results suggest that adding a group or shifting its position changes the potential of parental flavones and flavonols.

### 2.2. Acute Toxicity Study

The synthesized flavones (**F1–F6**) and flavonols (**OF1–OF6**) were evaluated for possible toxicological effects in mice and were found to be safe up to 650 mg.

### 2.3. Chemical Induced Stress

The synthetic flavonoids (flavones **F1–F6**) and (flavonols **OF1–OF6**) at a dose of 10 mg/kg showed a significant (*p* < 0.05, *p* < 0.01 and *p* < 0.001) reduction in the number of writhes within 20 min in acetic acid-induced stress mice when compared to stress in untreated mice.

The results from protection of chemical-induced writhes by flavones and flavonols suggested that the attenuation of stress produced by acetic acid (Table 1). Among the flavonoids; **OF3**, **OF2**, **F3** and **F2** showed a significant reduction in number of writhes and percentage of antistress response (7.03 ± 1.79, 80.94%, *n* = 8, *p* < 0.001), (12.19 ± 2.98, 74.27%, *n* = 8, *p* < 0.001), (11.18 ± 1.56, 74.40%, *n* = 8, *p* < 0.001) and (13.12 ± 2.21, 72.30%, *n* = 8, *p* < 0.001), respectively when compared to the control. Diazepam significantly reduced the stress induced by acetic acid in animals to 4.02 ± 1.87, 91.51%, *p* < 0.001, *n* = 8).

Other flavones, **F1**, **F4–F6,** also showed protection to 58.94%, 65.78%, 64.59% and 64.09%, respectively, but that was moderate in comparison to the results previously discussed. Similar results were observed by flavonols (**OF1**, **OF4–OF6**).

### 2.4. Swimming Endurance Test

As shown in Table 2, animals pre-treated for 15 days with flavones and flavonols (10 mg/kg) demonstrated a marked increase in swimming time and significantly reduced immobility time compared to the control group. Diazepam at 2 mg/kg also reduced the immobility time to a significant level in the positive control group of animals.

A significant difference between the treatment groups regarding the immobility time of synthesized flavones and flavonols during the swimming endurance test was observed from the results of one-way ANOVA, compared to the control group.

The increase in swimming time and decrease in the duration of immobility served as parameters for the anti-stress response of flavonoids. The methoxy and methyl containing flavones (**F3** and **F2**) and flavonols (**OF3** and **OF2**) were shown to possess a significant anti-stress response by decreasing the immobility to 2.88 ± 0.26 and 3.12 ± 0.38 min, respectively, for **F3** and **F2** and 2.33 ± 0.28 and 2.95 ± 0.37 min for **OF3** and **OF2, respectively,** of mice, compared to the control group. Other flavones (**F1, F4–F6**) and flavonols (**OF1, OF4–OF6**) also showed a decrease in immobility time, but that was moderate in comparison to methoxy- and methyl-containing flavonoids.

### 2.5. Restraint Stress

The stress caused a considerable raise in the weight of the adrenal glands and lessened the weight of the spleen and brain significantly, compared to the normal control (vehicle) group.

The stress caused marked changes in the weight of the spleen and brain, which were significantly reduced, and the weight of the adrenal gland was significantly increased in mice compared to the control group. In animals pre-treated for 15 days with flavones and flavonols (10 mg/kg), a marked suppression of the stress-induced changes in the weights of the spleen, brain and adrenal glands were observed. The results are shown in Table 3**.** The flavonoids at 25 mg/kg dose significantly (* *p* < 0.05, ** *p* < 0.01 and *** *p* < 0.001, *n* = 8) increased the weight of the spleen and decreased the weight of the adrenal gland, compared to the stress control group. 

### 2.6. Assessment of Biochemical Parameters and Biomarker Study

The restraint stress for mice adversely affects glucose levels in the blood and other biochemical parameters, and considerably increases blood glucose triglycerides and the total cholesterol level, compared with the control vehicle group (Table 4).

In animals pre-treated for 15 days with flavones and flavonols (10 mg/kg), a marked decrease in blood glucose levels, triglycerides and total cholesterol expression were observed, compared with the stress group. In the standard group, treated with diazepam (2 mg/kg), a significant reduction in the levels of biochemical parameters was observed, compared to that of the stress control group.

The restraint stress also significantly affected levels of many oxidative stress parameters in the brain, increased levels of lipid peroxidation (TBARS) and reduced the level of SOD and catalase significantly, compared with the control group. In animals pre-treated with flavonols and flavones (10 mg/kg) for fifteen days, a marked reduction in the levels of lipid peroxidation (TBARS) were observed and was significant (* *p* < 0.05, ** *p* < 0.01 and *** *p* < 0.001, *n* = 8) when compared to the stress control group (Table 5). The flavones and flavonols also significantly increased the levels of catalase and SOD when compared to the stress control group. Diazepam (2 mg/kg) lowered levels of lipid peroxidation (TBARS) and increased the levels of SOD and catalase.

## 3. Discussion

In this modern era, individuals in a given society are living in stressful conditions and as mentioned before, stress is a body survival response [20], forcing the body to bitterly adapt itself, both mentally and physically, to cope with such difficult circumstances. Extreme stress may lead to endocrine disorders, immunosuppression, depression and hypertension, as reported in a number of patients [10], not only affecting the quality of their lives but also creating socio-economic imbalances [20]. A number of drugs are currently in use to control stress and depression. However, they are mainly associated with side effects and severe toxicities [10]. Plant drugs are generally considered to be associated with low effects and flavonoids are the abundant compounds founds in almost all plants that have shown considerable biological potentials. As they are aromatic compounds, their activities can be enhanced by derivatization or shifting groups from ortho to meta or para positions and vice versa. Thus, in the present study, this idea has been utilized to suggest an effective drug for the management of stress through the derivatization of flavonoids, which are a natural raw material.

In the chemically-induced (acetic acid) stress model, the administration of inducer caused the hyperalgesic condition on the nociception path to produce an increased number of writhes, signifying the development of stress. The results of this stress model in the current study showed the decline in the number of writhes, indicating that synthetic flavonoids at a dose of 10 mg/kg can play a pivotal role in the pain inhibition to provide anti-stress action, as also reported by other researchers [8,21].

In the swimming endurance model, mice were forced to swim in water in a constrained area that prevented them from escaping, revealing symptoms reflecting extreme stressful conditions. It has been established that agents (drugs) with anti-stress activity reduce immobility time (or increase swimming endurance) [22]. The present study, using this model, revealed a decrease in immobility time, indicating that synthetic flavonoids, either flavones or flavonols, have anti-stress properties at a dose of 10 mg/kg body weight.

Under stressful conditions, biochemical parameters are increased via the hypothalamic-pituitary adrenal axis due to an increase in the corticosteroids and adrenocorticotropic hormone (ACTH) and to mobilize stored carbohydrates, fats and lipids to elevate blood glucose, triglycerides and cholesterol levels [8].

In the current study, when animals were exposed to restraint stress, the biochemical (total cholesterol, triglyceride blood glucose,) levels were elevated, reflecting similar results to a number of other studies [8,22,23]. The rise in total cholesterol, triglyceride and blood glucose levels induced by stress were significantly reduced by synthetic flavonoids, either flavones or flavonols, at a dose of 10 mg/kg b.w., indicating their anti-stress potential.

During stressful conditions, the nervous system is enormously susceptible to high levels of lipid peroxidation (TBARS) as an outcome of oxidative damage and oxygen tension and oxidizable substrates are high, as reported in other studies [8]. The synthetic flavones and flavonols significantly reduced the peroxidation level (TBARS), indicating their anti-stress potentials through the reduction of oxidative stress. The restraint stress also caused a depletion in catalase and SOD in the brain that were significantly reversed by the administration of synthetic flavones and flavonols to significantly increase the levels of catalase and SOD. 

The study also showed the promising reversal effects on the weights of vital organs (adrenal gland, spleen and brain) using the stress model, with similar findings being observed by other researchers [23,24].

## 4. Materials and Methods 

### 4.1. Chemicals and Animals

The synthesized flavones (**F1–F6**) and flavonols (**OF1–OF6**) used were synthesized and reported by our group previously [17,18,19]. The chemical structures of these compounds are given in Figure 2.

Chemicals and solvents such as ethanol, hydrogen peroxide, methanol, DPPH, ABTS, and Tocopherol were of E. Merck grade. Balb/C mice (19–23 g) were procured from the Veterinary research institute (VRI), Lahore and the National Institute of Health (NIH), Islamabad and kept at an animal house in standard plastic cages under standard laboratory conditions, with a temperature of 25 ± 2 °C, relative humidity of 55–65% and 12 h light/12 h dark cycle, with a standard diet and water ad libitum. Two weeks before the experiment, the animals were acclimatized to laboratory conditions. The animals were treated following the principles mentioned in the “Animals Byelaws 2008 of University of Malakand (Scientific Procedures Issue-I)”. Approval for this study was granted by the Ethical Committee of the Department of Pharmacy, in accordance with the Animals Byelaws 2008 of University of Malakand, vide notification no: Pharm/EC-HEREF/10-31/20.

### 4.2. In Vitro Antioxidant Activity

The antioxidant activity of compounds and standard tocopherol were assessed using DPPH (2,2-diphenyl-1-picrylhydrazyl) and ABTS (2,2′-azino-bis(3-ethylbenzothiazoline-6-sulfonate) models. For DPPH activity, synthetic flavones, flavonols and standard tocopherol at a concentration range of 6.25–50 μg/mL were treated with DPPH solution, and absorbance was recorded at 517 nm using a spectrophotometer Shimadzu UV-1800, Kyoto Japan.

Similarly, for ABTS activity, sample (0.1 mL) and standard solution at similar concentrations were mixed with ABTS and absorbance was measured at 734 nm. Scavenging activities for DPPH and ABTS were calculated and the IC_50_ was determined [25].

### 4.3. Acute Toxicity Study

The synthesized flavones (**F1–F6**) and flavonols (**OF1–OF6**), as illustrated in Figure 2, were subjected for possible toxicological effects in mice. The compounds at different dose concentrations were given in two phases and effects were noted. Similarly, preliminary pharmacological activity was also performed at various dose concentrations to determine the effective dose for anti-stress activity using various models [26].

### 4.4. Anti-Stress Activity

#### 4.4.1. Chemical Induced Stress

The anti-stress effect of synthesized flavonoids was screened for chemical-induced stress using the acetic acid model. Animals were randomly assigned into two groups of 8. The control group received Tween 80 (2%). Preliminary analysis was conducted to determine the effective dose using various dose concentrations (1–100 mg/kg body weight; b.w.) and then the animals in each group were treated with 10 mg/kg of respective flavonoids. Animals in the standard group received standard diazepam (2 mg/kg) through IP (intraperitoneal) route. The treatments in all groups were given persistently for a period of 15 days. On 15th day, the animals in each group received acetic acid i.p. at a dose of 0.1 mL (6% *v*/*v*) one hour after the drug treatment and the number of writhes was observed for 20 min [26].

#### 4.4.2. Swimming Endurance Test

Animals were divided at random into two groups of 8. Tween 80 (2%) was given to the control group, while animals in each group were treated with 10 mg/kg of each flavonoid. Animals in the standard group received diazepam (2 mg/kg) through IP route. All the doses were given constantly for 15 days. On 15th day, one hour after the dose administration, all the animals were allowed to swim in isolation in a glass tank. The time of immobility for each animal was recorded for a period of 30 min [26,27].

#### 4.4.3. Restraint Stress

In a similar fashion to both the above models for stress, after administration of samples and standard (10 mg/kg p.o and 2 mg/kg IP, respectively), stress was induced by tying the forelimbs and hind limbs of each animal in the respective group (*n* = 8), using adhesive tape for 2 h to immobilize them. The adhesive tape was removed, and blood samples were collected. After blood collection, the animals were sacrificed, and the brain and vital organs were isolated and weighed [26].

### 4.5. Assessment of Biochemical Parameters and Biomarker Study

After restrain test, blood samples were collected for the assessment of blood glucose, triglycerides and total cholesterol levels, using available commercial kits [26]. The isolated brain was used for the estimation of antioxidant enzymes catalase (CAT), glutathione (GSH) and MDA (Malondialdehyde) level for lipid peroxidation, using TBA (Thiobarbituric acid) reactive substances [28].

## 5. Conclusions

In the present study, synthetic derivatives of flavonoids were evaluated for their free radical scavenging potentials, using DPPH and ABTS assays. The para substituted methoxy derivatives were found to be the most potent agents against the tested free radicals. Significant antistress activities were also observed for these derivatives, determined through the most common predictive tests in use for screening stress in animal models, followed by a critical evaluation of the weight of vital organs and their biomarker levels. These synthetic flavones and flavonols require further in-depth investigation to explore their neuroprotective potential associated with oxidative stress in other animal models closely related to humans. These synthetic compounds may serve as potential candidates for anti-amnesic and antidepressant agents.

## Figures and Tables

**Figure 1 molecules-27-01402-f001:**
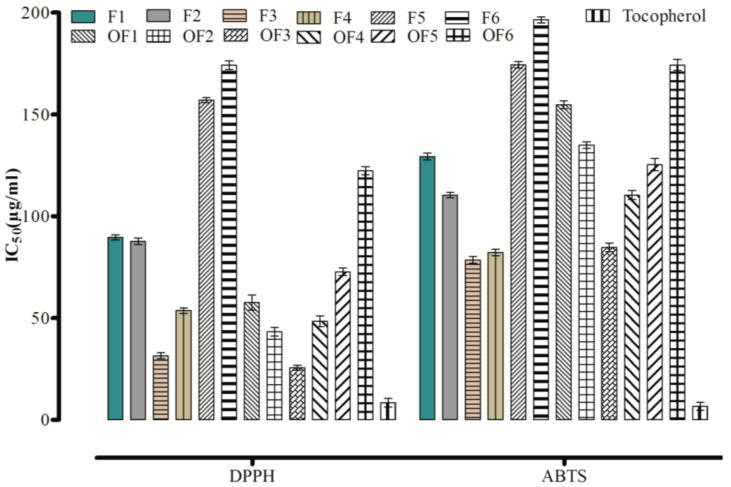
Antioxidant effects of flavones and flavonols.

**Figure 2 molecules-27-01402-f002:**
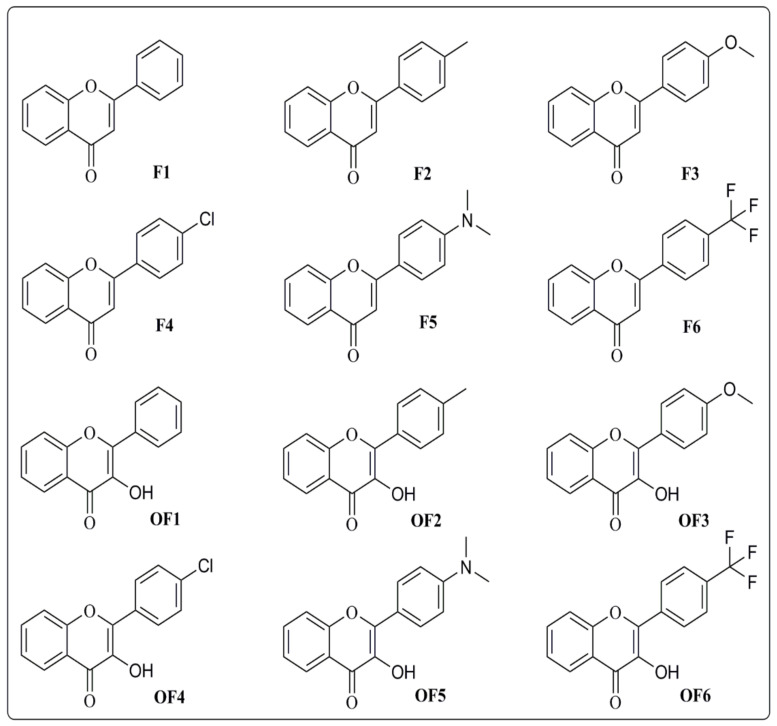
Synthetic compounds (**F1–F6**) and (**OF1–OF6**) used in the study.

**Table 1 molecules-27-01402-t001:** Effect of flavonoids on chemical induced stress test.

Group	Number of Writhes	Antistress Response (%)
Control (2% Tween80)	47.38 ± 2.66	-
**F1**	19.45 ± 1.89 *	58.94
**F2**	13.12 ± 2.21 ***	72.30
**F3**	11.18 ± 1.56 ***	74.40
**F4**	16.21 ± 2.09 **	65.78
**F5**	16.31 ± 2.58 **	64.57
**F6**	17.01 ± 1.89 **	64.09
**OF1**	17.32 ± 2.33 *	63.44
**OF2**	12.19 ± 2.98 ***	74.27
**OF3**	9.03 ± 1.79 ***	80.94
**OF4**	14.18 ± 2.17 **	67.07
**OF5**	16.44 ± 3.04 **	63.30
**OF6**	15.88 ± 2.11 **	66.48
Standard	4.02 ± 1.87 ***	91.51

Mean ± SEM. * *p* < 0.05, ** *p* < 0.01 and *** *p* < 0.001 vs. control.

**Table 2 molecules-27-01402-t002:** Effect of flavonoids on swimming endurance test.

Group	Immobility Time (Min)
Control (2% Tween80)	12.56 ± 1.31
**F1**	4.05 ± 0.45 **
**F2**	3.12 ± 0.38 **
**F3**	2.88 ± 0.26 ***
**F4**	3.21 ± 0.19 **
**F5**	3.64 ± 0.21 *
**F6**	3.41 ± 0.23 **
**OF1**	4.32 ± 0.31 *
**OF2**	2.95 ± 0.37 ***
**OF3**	2.33 ± 0.28 ***
**OF4**	3.44 ± 0.25 **
**OF5**	3.25 ± 0.29 **
**OF6**	3.34 ± 0.25 **
Standard	6.12 ± 0.47

Mean ± SEM. * *p* < 0.05, ** *p* < 0.01 and *** *p* < 0.001 vs. control.

**Table 3 molecules-27-01402-t003:** Effect of flavonoids on organ weights after restraint stress.

Group	Weight in mg		
	Adrenal Gland	Spleen	Brain
Control	5.45 ± 1.07	249.3 ± 2.21	453.3 ± 3.11
Stress control	15.02 ± 1.78 ^†††^	101.8 ± 2.56 ^†††^	364.7 ± 2.44 ^†††^
**F1**	8.13 ± 1.69 *	171.5 ± 2.71 *	398.1 ± 2.57 *
**F2**	7.63 ± 1.11 **	180.4 ± 2.38 *	401.6 ± 3.11 **
**F3**	7.22 ± 1.51 **	181.7 ± 2.06 **	411.3 ± 2.93 ***
**F4**	8.03 ± 1.98 *	173.1 ± 2.66 *	402.4 ± 3.19 **
**F5**	8.12 ± 1.81 *	170.3 ± 2.61 *	412.7 ± 2.81 **
**F6**	8.09 ± 1.77 *	171.1 ± 2.40 *	408.3 ± 2.60 **
**OF1**	8.09 ± 1.34 *	188.4 ± 2.24 **	397.4 ± 3.76
**OF2**	7.21 ± 1.87 ***	190.5 ± 2.76 **	414.2 ± 2.51 ***
**OF3**	6.79 ± 1.10 ***	194.8 ± 2.11 ***	413.6 ± 2.66 ***
**OF4**	7.91 ± 1.91 *	183.9 ± 2.56 **	401.8 ± 2.34 **
**OF5**	8.04 ± 1.18 *	186.3 ± 2.74 **	400.5 ± 2.21 *
**OF6**	8.11 ± 1.31 *	179.1 ± 2.30 **	403.9 ± 2.239 *
Standard	10.58 ± 5.77 ^ns^	158.3 ± 2.89 *	390.8 ± 2.75 *

Mean ± SEM. * *p* < 0.05, ** *p* < 0.01 and *** *p* < 0.001 vs. stress control. **^†††^**
*p* < 0.001 vs. control. ns; non-significant.

**Table 4 molecules-27-01402-t004:** Effect of flavonoids on different biochemical parameters in blood after restraint stress.

Group	Biochemical Parameters (Blood)		
	Glucose	TGs	Total Ch
Control	96.43 ± 1.89	52.31 ± 0.98	79.05 ± 0.54
Stress control	139.62 ± 2.24 ^†††^	101.22 ± 1.34 ^†††^	141.72 ± 1.75 ^†††^
**F1**	105.22 ± 2.15 ^ns^	54.39 ± 0.87 *	81.33 ± 0.61 *
**F2**	99.23 ± 1.69 *	45.09 ± 0.67 ***	78.61 ± 0.78 **
**F3**	98.45 ± 1.51 **	44.69 ± 0.37 ***	78.43 ± 0.93 ***
**F4**	100.92 ± 1.91 *	47.51 ± 0.38 ***	80.76 ± 0.91 **
**F5**	101.33 ± 1.72 *	46.72 ± 0.51 ***	81.82 ± 0.85 *
**F6**	102.09 ± 1.62 *	47.26 ± 0.44 ***	82.01 ± 0.79 *
**OF1**	104.55 ± 1.77 ^ns^	51.22 ± 0.66 **	80.81 ± 0.72 **
**OF2**	97.56 ± 1.61 ***	42.88 ± 0.41 ***	76.82 ± 0.86 ***
**OF3**	97.06 ± 1.71 ***	43.11 ± 0.36 ***	76.09 ± 0.64 ***
**OF4**	101.15 ± 1.66 *	46.92 ± 0.44 ***	78.33 ± 0.76 **
**OF5**	101.09 ± 1.14 *	46.24 ± 0.39 ***	77.91 ± 0.97 ***
**OF6**	100.32 ± 1.29 *	44.02 ± 0.30 ***	77.02 ± 0.97 ***
Standard	108.22 ± 1.06 ^ns^	38.52 ± 0.57 ***	75.62 ± 0.65 ***

Mean ± SEM. * *p* < 0.05, ** *p* < 0.01 and *** *p* < 0.001 vs. stress control. **^†††^**
*p* < 0.001 vs. control. ns; non-significant.

**Table 5 molecules-27-01402-t005:** Effect of flavonoids on different biochemical parameters in brain after restraint stress.

Group	Biochemical Parameters (Brain)		
	Catalase (IU/dL)	SOD (IU/dL)	TBARS (nmol/g)
Control	23.65 ± 0.89	15.75 ± 0.58	14.21 ± 0.41
Stress control	6.71 ± 0.46 ^†††^	4.11 ± 0.54 ^†††^	27.92 ± 0.75 ^†††^
**F1**	16.33 ± 0.69 ***	7.96 ± 0.60 ***	19.42 ± 0.51 ***
**F2**	18.21 ± 0.61	8.77 ± 0.59	18.72 ± 0.62 ***
**F3**	18.92 ± 0.51 ***	9.42 ± 0.87 ***	18.44 ± 0.63 ***
**F4**	17.88 ± 0.59 ***	9.31 ± 0.78 ***	19.09 ± 0.49 ***
**F5**	17.11 ± 0.51 ***	8.04 ± 0.57 ***	19.02 ± 0.71 **
**F6**	17.36 ± 0.62 ***	8.13 ± 0.51 ***	18.96 ± 0.60 **
**OF1**	17.02 ± 0.71 ***	8.46 ± 0.76 ***	19.25 ± 0.51 ***
**OF2**	18.74 ± 0.67	9.02 ± 0.54	17.74 ± 0.56 ***
**OF3**	19.02 ± 0.77 ***	9.58 ± 0.66 ***	17.49 ± 0.61 ***
**OF4**	17.45 ± 0.66 ***	8.35 ± 0.64 ***	18.67 ± 0.76 ***
**OF5**	17.06 ± 0.61 ***	8.11 ± 0.51 ***	18.56 ± 0.51 *
**OF6**	17.01 ± 0.52 ***	8.03 ± 0.49 ***	18.10 ± 0.51 *
Standard	17.25 ± 0.66 ***	6.56 ± 0.57 ***	18.22 ± 0.65 ***

Mean ± SEM. * *p* < 0.05, ** *p* < 0.01 and *** *p* < 0.001 vs. stress control. **^†††^**
*p* < 0.001 vs. control.

## Data Availability

Not applicable.

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
