# Peer review of "In Vivo Antistress Effects of Synthetic Flavonoids in Mice: Behavioral and Biochemical Approach"

_molecules, 2022, doi:10.3390/molecules27041402_

Round 1

Reviewer 1 Report

The manuscript “In-vivo antistress effects of synthetic flavonoids in mice: behavioral and biochemical approach” describes antioxidant bioassays and in vivo anti-stress tests for flavones and flavonoids. The manuscript deserves publication after major revision.

Suggestions for the authors to consider are listed below.

Line 20: … in-vivo

Suggestion: … in vivo

Please correct all over the manuscript.

Line 22: … the mono par substituted methoxy containing F3 and OF3 produced …

Suggestion: … the mono para substituted methoxy containing F3 and OF3 produced …

“para” must be italic.

Line 24: … μg/ml and 25.54±1.21 μg/ml respectively.

Suggestion: … μg/mL and 25.54 ± 1.21 μg/mL respectively.

mL not ml all over the manuscript. “Replace all” tool in MS Word can be used to do this.

Line 29: … reduction in number of writhes and increase in swimming endurance time respectively.

Suggestion: … reduction in number of writhes and increase in swimming endurance time.

Line 33: … when compare to stress group.

Suggestion: … when compared to stress group.

Line 33: … Significant rise in the level of catalase and SOD was also observed.

Suggestion: Please state the meaning of the acronym SOD.

Line 37: … against a diverse physiological and biochemical alteration …

Suggestion: … against diverse physiological and biochemical alterations …

Line 46: … that about 450 million population is suffering from psychiatric disorders …

Suggestion: … that about 450 million people are suffering from psychiatric disorders …

Line 47: During stressful conditions like acute restraint stress (ARS) stimulates many cellular events and increased the body energy requirements, …

Suggestion: During stressful conditions like acute restraint stress (ARS) many cellular events are stimulated, and body energy requirements are increased, …

Line 50: These reactive free radicals cause damages to many parts of body and specially CNS due to brain high consumption of oxygen, …

Suggestion: These reactive free radicals cause damages to many parts of the body and specially of CNS due to brain high consumption of oxygen, …

Please state the meaning of the acronym CNS.

Lines 52-54: Likewise, in rodents, the restraint stress triggers numerous hormonal, neurochemical, and behavioral dysfunctions that are frequently linked with an imbalance of intracellular redox state in the brain.

Suggestion: Likewise, in rodents, restraint stress triggers numerous hormonal, neurochemical, and behavioral dysfunctions that are often associated with an imbalance of the intracellular redox state in the brain.

Lines 109: In-vitro antioxidant activity

Suggestion: In vitro antioxidant activity

Please correct all over the manuscript.

Lines 71-72: Flavonoids are a group of secondary metabolites that are extensively present in plant kingdom and have been recognized for their exciting medicinal actions

Suggestion: Flavonoids belong to a group of secondary metabolites that are widely present in the plant kingdom and have been recognized for their exciting medicinal actions.

Line 102: … and 12hour light/12 h dark cycle

Suggestion: … and 12 h light/12 h dark cycle

Lines 117: Scavenging activity for DPPH and ABTS was calculated and the IC50 was determined [20].

Suggestion: Scavenging activities for DPPH and ABTS were calculated and the IC50 were determined [20].

Lines 139: … each group were treated with 10 mg/kg of respective flavonoids.

Suggestion: … each group were treated with 10 mg/kg of each flavonoid.

Lines 140: i.p. route.

Suggestion: What is i.p. route?

Lines 162: … the mono par substituted methoxy containing …

Suggestion: … the mono para substituted methoxy containing …

Lines 169: It is evident that mono par substituted methoxy containing flavone …

Suggestion: It is evident that mono para substituted methoxy containing flavone …

Lines 169: These results propose that adding the group or shifting their position may probably increase or decrease the potentiality of flavones and flavonols.

Suggestion: These results suggest that adding a group or shifting its position changes the potential of flavones and flavonols.

Question: This is not obvious? A different flavonone may not present different potential?

Lines 246-269: The stressful social conditions are one of the primary features among modern society of an individual life style [25] and stress is a survival response that toughen the mental and physical status. The extreme stress may lead to endocrine disorders, immunosuppression, depression and hypertension increased number of patients [10] affecting the quality of life and socio-economic imbalance [25]. A number of drugs are currently used to control stress and depression but are associated with side effects and severe toxicity [10]. Thus, management of stress through synthetic flavonoids related to natural source may be a valuable alternative to anti-stress drugs.

In chemical induced (acetic acid) stress model, the administration of inducer caused the hyperalgesic condition on the nocicception path to produce the increase number of writhes signifying the development of stress. The results of this stress model in the current study show the decline in the number of writhes indicating the synthetic flavonoids at a dose of 10 mg/kg can play a pivotal role in the pain inhibition to have its anti-stress action like several other reports [8,26].

In swimming endurance model, the animal is strained to swim in water in a constrained area from where they cannot getaway reveals to the symptoms symbolize the extreme stress condition. It has been established that agents (drugs) with anti-stress activity reduces the immobility time (or increases the swimming endurance) and has been reported [27]. The present study using this model has revealed that there is decreases in the immobility time which indicates evidently that synthetic flavonoids either flavones or flavonols have anti-stress properties at a dose of 10 mg/kg.

Under the stressful conditions, the biochemical parameters get increased via hypothalamic-pituitary adrenal axis by increasing corticosteroids and adrenocorticotropic hormone (ACTH) and to mobilize the stored carbohydrates, fats and lipids to elevate theblood glucose, triglycerides and cholesterol levels [8].

Suggestion: These paragraphs are not discussion. These look like introduction of the topic.

Conclusion: In conclusion, the present investigation indicates that synthetic flavones and flavonols has significant antistress activity …

Suggestion: The present investigation indicates that synthetic flavones and flavonols have significant antistress activities …

Question: This is a very succinct conclusion.

Author Response

Reviewer 1

The manuscript “In-vivo antistress effects of synthetic flavonoids in mice: behavioral and biochemical approach” describes antioxidant bioassays and in vivo anti-stress tests for flavones and flavonoids. The manuscript deserves publication after major revision.

Suggestions for the authors to consider are listed below.

Line 20: … in-vivo …

Suggestion: … in vivo …

Please correct all over the manuscript.

  • Thank you, worthy reviewer, it was corrected accordingly

Line 22: … the mono par substituted methoxy containing F3 and OF3 produced …

Suggestion: … the mono para substituted methoxy containing F3 and OF3 produced …

“para” must be italic.

  • Corrected accordingly

Line 24: … μg/ml and 25.54±1.21 μg/ml respectively.

Suggestion: … μg/mL and 25.54 ± 1.21 μg/mL respectively.

mL not ml all over the manuscript. “Replace all” tool in MS Word can be used to do this.

  • Thank you, worthy reviewer, it was corrected accordingly

Line 29: … reduction in number of writhes and increase in swimming endurance time respectively.

Suggestion: … reduction in number of writhes and increase in swimming endurance time.

  • Thank you, worthy reviewer, the sentence was corrected accordingly

Line 33: … when compare to stress group.

Suggestion: … when compared to stress group.

  • Corrected accordingly

Line 33: … Significant rise in the level of catalase and SOD was also observed.

Suggestion: Please state the meaning of the acronym SOD.

  • Defined accordingly

Line 37: … against a diverse physiological and biochemical alteration …

Suggestion: … against diverse physiological and biochemical alterations …

  • Thank you worthy reviewer, corrected accordingly

Line 46: … that about 450 million population is suffering from psychiatric disorders …

Suggestion: … that about 450 million people are suffering from psychiatric disorders …

  • Thank you worthy reviewer, corrected accordingly

Line 47: During stressful conditions like acute restraint stress (ARS) stimulates many cellular events and increased the body energy requirements, …

Suggestion: During stressful conditions like acute restraint stress (ARS) many cellular events are stimulated, and body energy requirements are increased, …

  • Thank you worthy reviewer, corrected accordingly

Line 50: These reactive free radicals cause damages to many parts of body and specially CNS due to brain high consumption of oxygen, …

Suggestion: These reactive free radicals cause damages to many parts of the body and specially of CNS due to brain high consumption of oxygen, …

  • Thank you worthy reviewer, corrected accordingly

Please state the meaning of the acronym CNS.

  • Defined accordingly

Lines 52-54: Likewise, in rodents, the restraint stress triggers numerous hormonal, neurochemical, and behavioral dysfunctions that are frequently linked with an imbalance of intracellular redox state in the brain.

Suggestion: Likewise, in rodents, restraint stress triggers numerous hormonal, neurochemical, and behavioral dysfunctions that are often associated with an imbalance of the intracellular redox state in the brain.

  • Thank you worthy reviewer, corrected accordingly

Lines 109: In-vitro antioxidant activity

SuggestionIn vitro antioxidant activity

Please correct all over the manuscript.

  • Thank you worthy reviewer, corrected accordingly in whole manuscript

Lines 71-72: Flavonoids are a group of secondary metabolites that are extensively present in plant kingdom and have been recognized for their exciting medicinal actions

Suggestion: Flavonoids belong to a group of secondary metabolites that are widely present in the plant kingdom and have been recognized for their exciting medicinal actions.

  • Thank you worthy reviewer, corrected accordingly

Line 102: … and 12hour light/12 h dark cycle

Suggestion: … and 12 h light/12 h dark cycle

  • Thank you worthy reviewer, corrected accordingly

Lines 117: Scavenging activity for DPPH and ABTS was calculated and the IC50 was determined [20].

Suggestion: Scavenging activities for DPPH and ABTS were calculated and the IC50 were determined [20].

  • Thank you worthy reviewer, corrected accordingly

Lines 139: … each group were treated with 10 mg/kg of respective flavonoids.

Suggestion: … each group were treated with 10 mg/kg of each flavonoid.

  • Thank you worthy reviewer, corrected accordingly

Lines 140: i.p. route.

Suggestion: What is i.p. route?

  • Defined accordingly

Lines 162: … the mono par substituted methoxy containing …

Suggestion: … the mono para substituted methoxy containing …

  • Thank you worthy reviewer, corrected accordingly

Lines 169: It is evident that mono par substituted methoxy containing flavone …

Suggestion: It is evident that mono para substituted methoxy containing flavone …

  • Thank you, worthy reviewer, corrected accordingly

Lines 169: These results propose that adding the group or shifting their position may probably increase or decrease the potentiality of flavones and flavonols.

Suggestion: These results suggest that adding a group or shifting its position changes the potential of flavones and flavonols.

  • Thank you worthy reviewer, corrected accordingly

Question: This is not obvious? A different flavonone may not present different potential?

  • Worthy reviewer, as you know it better those compounds involving benzene rings on substitution are activated in some cases whereas in other cases, they deactivated due to electron donating or withdrawing effects of the group. Similarly, shifting groups from ortho to meta or para position on benzene ring alters reactivity and consequently its biological potentials.

Lines 246-269: The stressful social conditions are one of the primary features among modern society of an individual life style [25] and stress is a survival response that toughen the mental and physical status. The extreme stress may lead to endocrine disorders, immunosuppression, depression and hypertension increased number of patients [10] affecting the quality of life and socio-economic imbalance [25]. A number of drugs are currently used to control stress and depression but are associated with side effects and severe toxicity [10]. Thus, management of stress through synthetic flavonoids related to natural source may be a valuable alternative to anti-stress drugs.

In chemical induced (acetic acid) stress model, the administration of inducer caused the hyperalgesic condition on the nocicception path to produce the increase number of writhes signifying the development of stress. The results of this stress model in the current study show the decline in the number of writhes indicating the synthetic flavonoids at a dose of 10 mg/kg can play a pivotal role in the pain inhibition to have its anti-stress action like several other reports [8,26].

In swimming endurance model, the animal is strained to swim in water in a constrained area from where they cannot getaway reveals to the symptoms symbolize the extreme stress condition. It has been established that agents (drugs) with anti-stress activity reduces the immobility time (or increases the swimming endurance) and has been reported [27]. The present study using this model has revealed that there is decreases in the immobility time which indicates evidently that synthetic flavonoids either flavones or flavonols have anti-stress properties at a dose of 10 mg/kg.

Under the stressful conditions, the biochemical parameters get increased via hypothalamic-pituitary adrenal axis by increasing corticosteroids and adrenocorticotropic hormone (ACTH) and to mobilize the stored carbohydrates, fats and lipids to elevate the blood glucose, triglycerides and cholesterol levels [8].

Suggestion: These paragraphs are not discussion. These look like introduction of the topic.

  • Worthy reviewer, chemical and physical methods have been used to induce stress in experimental animals the explanation given here is related to the experiment performed showing similarity of our study with those reported in literature and are therefore mandatory to be place over here. Also, if these are shifting the discussion will look like a short summary.

Conclusion: In conclusion, the present investigation indicates that synthetic flavones and flavonols has significant antistress activity …

Suggestion: The present investigation indicates that synthetic flavones and flavonols have significant antistress activities …

Question: This is a very succinct conclusion.

  • Worthy reviewer, the conclusion was accordingly elaborated. Hopefully, it will be ok now.

Reviewer 2 Report

Dear authors,

The draft paper you proposed reports a series of synthetic flavonoids and the corresponding antistress activity in an in-vivo mouse model.

While congratulating the authors for the meaningful project carried out, this reviewer believes that a series of issues should be fixed prior to publication, in order to have a good final version of your report. 

  • The abstract is too long and distracting. Consider re-write it shorter and clearer. 
  • Figure 1 is not of very good quality; it should be improved. 
  • DPPH and ABTS, even if they are clearly known acronyms, should be fully named at least the first time they are presented, or an acronyms list should be inserted in the paper to improve the clarity. 
  • Every now and then in the text, some bold sentences can be found. Is it intended or a typing error? If it is intended, is it necessary? See as an example row 70 or 200.
  • Compounds' labels, such as F1-F6 or OF1-OF6 are sometimes reported in bold, sometimes not. They should be uniformed. 

Author Response

Reviewer 2:

The draft paper you proposed reports a series of synthetic flavonoids and the corresponding antistress activity in an in-vivo mouse model.

While congratulating the authors for the meaningful project carried out, this reviewer believes that a series of issues should be fixed prior to publication, in order to have a good final version of your report. 

  • The abstract is too long and distracting. Consider re-write it shorter and clearer. 
  • The abstract was rephrased accordingly.
  • Figure 1 is not of very good quality; it should be improved. 
  • The figure was replaced accordingly.
  • DPPH and ABTS, even if they are clearly known acronyms, should be fully named at least the first time they are presented, or an acronyms list should be inserted in the paper to improve the clarity. 
  • Defined accordingly
  • Every now and then in the text, some bold sentences can be found. Is it intended or a typing error? If it is intended, is it necessary? See as an example row 70 or 200.
  • Worthy reviewer, thank you for the suggestion, they were corrected accordingly
  • Compounds' labels, such as F1-F6 or OF1-OF6 are sometimes reported in bold, sometimes not. They should be uniformed. 
  • Worthy reviewer, thank you for the suggestion, they were corrected accordingly

Round 2

Reviewer 1 Report

The manuscript “In-vivo antistress effects of synthetic flavonoids in mice: behavioral and biochemical approach” describes antioxidant bioassays and in vivo anti-stress tests for flavones and flavonoids. The manuscript deserves publication after revision.

Suggestions for the authors to consider are listed below.

Title: … in-vivo

Suggestion: … in vivo

Please correct all over the manuscript.

Line 22: … the mono para substituted methoxy containing F3 and OF3 produced …

Suggestion: “para” must be italic.

Line 24: … μg/mL and 25.54±1.21 μg/mL

Suggestion: … 25.54 ± 1.21 μg/mL (add space between numbers and ±).

Repeat this operation all over the manuscript.

Line 57: These reactive free radicals cause damages to many parts of the body, especially to CNS (central nervous system) as in CNS, brain have; high consumption of oxygen, …

Suggestion: … especially to CNS (central nervous system) as in CNS, brain have; …

“This part of the sentence does not make sense”. What is “brain have”? What is “CNS as in CNS”?

Lines 166: In-vitro antioxidant activity

Suggestion: In vitro antioxidant activity

Please correct all over the manuscript.

Lines 126: Scavenging activity for DPPH and ABTS was calculated and the IC50 was determined [20].

Suggestion: Scavenging activities for DPPH and ABTS were calculated and the IC50 were determined [20].

Lines 142: i.p. route.

Suggestion: What is i.p. route?

Lines 172: … the mono para substituted methoxy containing …

Suggestion: “para” must be in italic.

Lines 169: It is evident that mono para substituted methoxy containing flavone …

Suggestion: “para” must be in italic.

Conclusion: The para substituted methoxy derivatives

Suggestion: “para” must be in italic.

Author Response

Reviewer 1:

The manuscript “In-vivo antistress effects of synthetic flavonoids in mice: behavioral and biochemical approach” describes antioxidant bioassays and in vivo anti-stress tests for flavones and flavonoids. The manuscript deserves publication after revision.

Suggestions for the authors to consider are listed below.

Title: … in-vivo …

Suggestion: … in vivo …

Please correct all over the manuscript.

  • Worthy reviewer, thank you it was accordingly corrected in the whole manuscript

Line 22: … the mono para substituted methoxy containing F3 and OF3 produced …

Suggestion: “para” must be italic.

  • Corrected accordingly

Line 24: … μg/mL and 25.54±1.21 μg/mL

Suggestion: … 25.54 ± 1.21 μg/mL (add space between numbers and ±).

Repeat this operation all over the manuscript.

  • Corrected accordingly

Line 57: These reactive free radicals cause damages to many parts of the body, especially to CNS (central nervous system) as in CNS, brain have; high consumption of oxygen, …

Suggestion: … especially to CNS (central nervous system) as in CNS, brain have; …

“This part of the sentence does not make sense”. What is “brain have”? What is “CNS as in CNS”?

  • The sentence was rephrased accordingly

Lines 166: In-vitro antioxidant activity

SuggestionIn vitro antioxidant activity

Please correct all over the manuscript.

  • Corrected accordingly

Lines 126: Scavenging activity for DPPH and ABTS was calculated and the IC50 was determined [20].

Suggestion: Scavenging activities for DPPH and ABTS were calculated and the IC50 were determined [20].

  • Rephrased accordingly

Lines 142: i.p. route.

Suggestion: What is i.p. route?

  • (intraperitoneal). It is already defined there

Lines 172: … the mono para substituted methoxy containing …

Suggestion: “para” must be in italic.

  • Corrected accordingly

Lines 169: It is evident that mono para substituted methoxy containing flavone …

Suggestion: “para” must be in italic.

  • Corrected accordingly
  •  

Conclusion: The para substituted methoxy derivatives

Suggestion: “para” must be in italic.

  • Corrected accordingly